# Assessing the Validity and Reliability of a Single Lumbar-Mounted IMU System for Gait Analysis

**DOI:** 10.3390/s25247643

**Published:** 2025-12-17

**Authors:** Alfredo Lerín-Calvo, Giuseppe Prisco, Elena Fernández-Maza, Marta Núñez-González, Gema Santiago-Lorrio, Leandro Donisi, Sergio Lerma-Lara

**Affiliations:** 1Grupo de Investigación de Neurociencias Aplicadas a la Rehabilitación (GINARE), 28293 Alcorcón, Spain; 2Grupo de Investigación Clínico Docente Sobre Ciencias de la Rehabilitación (INDOCLIN), CSEU La Salle, UAM, Aravaca, 28023 Madrid, Spain; 3Department of Medicine and Health Sciences, University of Molise, 86100 Campobasso, Italy; g.prisco2@studenti.unimol.it; 4Department of Physiotherapy, Centro Superior de Estudios Universitarios La Salle, Universidad Autónoma de Madrid, Aravaca, 28023 Madrid, Spain; elena.ferma@hotmail.com (E.F.-M.); nunezgonzalez.marta10@gmail.com (M.N.-G.);; 5Department of Advanced Medical and Surgical Sciences, University of Campania Luigi Vanvitelli, 80138 Naples, Italy

**Keywords:** gait analysis, inertial measurement unit, motion capture, reliability, spatiotemporal parameters, validity

## Abstract

Wearable inertial sensors offer a practical alternative to gold-standard optoelectronic systems for gait analysis, though their validity remains uncertain due to sensor placement. This study examined the intra-rater and inter-rater reliability and the concurrent validity of a single lumbar-mounted inertial measurement unit (Baiobit, BTS Bioengineering, Garbagnate Milanese, Italy) compared with a 3D optoelectronic system (SMART-DX 6000, BTS Bioengineering). Thirty healthy adults walked along an 8 m walkway at a self-selected speed, and seven spatiotemporal gait parameters (cadence, velocity, stride length, step length, stance, swing, and single-support phases) were computed by both systems. Reliability and validity were assessed using intraclass correlation coefficients (ICC), standard error of measurement, minimum detectable change, paired tests, Spearman correlation, Passing–Bablok regression, and Bland–Altman analysis. Baiobit showed intra- and inter-rater ICCs of 0.53–0.90 and 0.66–0.88, respectively. Bland–Altman results indicated non-significant biases for global parameters (velocity: −0.06 m/s; cadence: 1.11 steps/min), whereas spatial measures showed significant biases (stride length: 0.11 m; step length: 0.06 m). Gait phase parameters demonstrated low correlations (r = 0.08–0.11) and proportional systematic errors. Overall, the Baiobit system provided reliable and valid estimates of global spatiotemporal parameters but lacked precision for gait phase metrics, underscoring limitations that currently prevent it from fully replacing optoelectronic systems.

## 1. Introduction

Gait analysis is a fundamental tool for studying human movement, with broad applications in sports, clinical diagnostics, and rehabilitation [1,2,3,4,5,6]. Human locomotion is a highly coordinated process enabling forward progression with minimal energy expenditure. It is typically described as a repeated gait cycle composed of stance, swing, and double support phases and characterized by spatiotemporal and kinematic parameters that provide insights into joint motion and coordination [7]. Traditionally, quantitative gait analysis has relied on three-dimensional (3D) optoelectronic motion capture systems, widely regarded as the gold standard for their high precision and detailed kinematic output [8,9]. However, their use is constrained by high costs, technical complexity, and the need for dedicated laboratory environments, which limit their practicality in routine clinical practice and field-based applications [10]. In recent years, wearable technologies, particularly inertial measurement units (IMUs) combining accelerometers, gyroscopes, and magnetometers have emerged as a promising alternative to laboratory-based systems [1,11,12,13,14,15]. Compact, affordable, and portable, IMUs enable gait assessment in real-world environments outside the laboratory, providing more realistic evaluations of human movement and allowing for monitoring gait in individuals with neurological or musculoskeletal disorders to evaluating athletic performance [1,16,17,18,19]. However, limitations include drift, which can progressively compromise the accuracy of motion tracking, and sensitivity to sensor placement and signal choice, affecting the measurement consistency and reliability [20,21].

Existing studies on gait analysis using IMUs to estimate spatiotemporal gait parameters are highly varied and difficult to compare due to the differences in sensor placement, number of sensors used, and the type of inertial signals selected [21]. For instance, Teuf et al. [22] investigated the agreement between a seven-sensor IMU system (mounted on the shanks, feet, thighs, and lower back) and an optoelectronic system for spatiotemporal gait analysis. Their results indicated high validity for most parameters, with the exception of step width and swing width. Cimolin et al. [23] examined the validity of spatiotemporal parameter estimation using a single lower-trunk IMU in both obese and normal-weight adolescents. The authors reported that the inertial system was effective in capturing spatiotemporal parameters, though caution was advised when interpreting gait phase metrics. Zago et al. [24] compared a single lower-back IMU with an optoelectronic system, concluding that the two approaches were largely comparable, except for discrepancies in velocity. Digo et al. [25] evaluated three IMU configurations (trunk, shank, and ankle) compared with an optoelectronic system in a healthy elderly cohort. All setups performed well, although the trunk-mounted IMU consistently outperformed the others. Ricciardi et al. [26] assessed an inertial system based on three-IMU configuration (lower back and feet) against an optoelectronic reference in patients with Parkinson disease. Their findings showed constant systematic errors for cadence and gait cycle time, and proportional errors for gait phase parameters, limiting interchangeability between systems. Ziagkas et al. [27] evaluated the agreement between an inertial platform insole and an optoelectronic system in estimating kinematic and spatiotemporal parameters. The authors concluded that the inertial platform insole provided accurate measurements for temporal gait parameters but not for spatial parameters. Finally, Saggio et al. [28] evaluated the agreement of an inertial system based on seven-IMU configuration (lower back, shanks, feet, and thighs), reporting excellent agreement with the reference system for most spatiotemporal parameters, except step length and double support.

Within this growing landscape of wearable technologies, the Baiobit inertial system (Rivelo Srl, BTS Bioengineering Group, Milan, Italy) exemplifies this new generation of wearable devices in the field of gait analysis. The system has been validated against the GAITRite system (CIR Systems Inc., Franklin, NJ, USA) for estimating mean spatiotemporal parameters, showing high validity for velocity, cadence, and step and stride length, but lower validity for stance and swing times [29]. While these findings are promising, Baiobit system accuracy in estimating spatiotemporal parameters compared with gold-standard 3D motion capture systems has not yet been established. Validation is essential to determine whether Baiobit can serve as a reliable, practical alternative to laboratory-based systems. A single lumbar IMU offers methodological advantages, including minimal setup time, ease of use, and the ability to capture gait without restricting natural movement. Such a configuration is particularly suitable for clinical scenarios where rapid assessment is needed, including primary care triage, rehabilitation follow-up, and community-based gait screening. Therefore, this study aimed to evaluate the intra- and inter-rater reliability, as well as the concurrent validity, of the Baiobit system compared with a 3D optoelectronic motion capture system for spatiotemporal gait parameters in healthy adults. We hypothesized that the single lumbar IMU would demonstrate good-to-excellent agreement with the reference system and acceptable test–retest reliability.

## 2. Materials and Methods

### 2.1. Instrumentation

#### 2.1.1. Three-Dimensional Optoelectronic System

BTS SMART-DX 6000 (BTS Bioengineering, Garbagnate Milanese, Italy) is a 3D optoelectronic motion capture system designed for accurate biomechanical analysis (Figure 1a). It is widely utilized in physiotherapy, rehabilitation and sports science applications [7]. The system includes eight infrared cameras operating at a sampling frequency of 200 Hz, two dynamometric platforms, and twenty-two retro-reflective passive markers (each one with a diameter of 14 mm). Data acquisition and processing are managed through the BTS SMART-Clinic software platform (SMARTClinic v1.10; BTS Bioengineering, Garbagnate Milanese, Italy). Previous studies have demonstrated the reliability of the BTS system in the assessment of gait parameters, indicating strong measurement consistency [30].

#### 2.1.2. Wearable Inertial System

The Baiobit system is a commercial wearable inertial system designed for human motion analysis (Figure 1b). The system is composed of IMUs operating at 200 Hz which integrate a triaxial accelerometer (16-bit, 4–1000 Hz, ranges of ±16 g), a triaxial gyroscope (16-bit, 4–8000 Hz, ranges of ±2000 °/s), and a triaxial magnetometer (13-bit, 100 Hz, range of ±1200 uT). The system also includes dedicated software for collecting, analyzing, and storing kinematic data. The Baiobit system has been widely applied in gait assessment studies [11,29,31].

### 2.2. Study Design and Study Population

This prospective study employed a single-group repeated-measures design to assess the inter- and intra-rater reliability and validity of the Baiobit system compared with the BTS SMART-DX 6000. Two raters (A and B) independently performed all measurements used to evaluate inter-rater reliability. To minimize potential order effects, the sequence in which each rater conducted the assessments was randomized using a 1:1 block randomization scheme. A third rater (C), who was not involved in performing the measurements, was responsible for maintaining blinding between the two raters and ensuring that neither had access to the other’s results. Rater C also oversaw the collection, verification, and compilation of the measurement data obtained from both raters.

The sample size of study population was calculated using the Intraclass Correlation Coefficient (ICC) method [32,33], targeting an ICC of 0.80 with a significance level of 0.05 and 80% statistical power (β = 0.20), assuming a null hypothesis ICC of 0.60. A minimum of 27 participants was required; accounting for a 10% expected dropout, the final target was 30. Accordingly, 30 healthy individuals (16 females and 14 males) were recruited in the present study. Of these, 28 were included in the validity analysis and 29 in the reliability analysis.

The study was approved by the ethical committee of La Salle University Centre, and an informed consent was obtained from all participants (CSEULS-PI-011/2025). Additionally, the study adhered to recommendations from COSMIN and STARD guidelines for reliability and validity research, ensuring standardized reporting of measurement properties. The demographic characteristics of the study population, reported as mean ± standard deviation (SD), are presented in Table 1.

### 2.3. Sensor Equipment and Data Acquisition

Gait analysis tasks were performed in the movement analysis laboratory at the La Salle University Hospital. Before data collection, anthropometric measurements were recorded for each participant according to the protocol described by Collins et al. [34] and the BTS SMART-DX 6000 system was calibrated to ensure optimal capture range. Twenty-two passive reflective markers were placed on specific anatomical landmarks according to Helen Hayes model with medial markers [34]. Particularly, three markers were positioned on the trunk, three on the pelvis, two on each thigh, two on each tibia, two on each foot, and one medial marker on each thigh and tibia. Additionally, a single Baiobit sensor was placed on the lower back at the level of the second sacral vertebra (S2) using an adaptive belt. Since the reflected marker and the Baiobit sensor shared the same body location at S2, the marker was placed directly on the sensor, enabling simultaneous data recording by both the Baiobit sensor and the BTS optoelectronic system (Figure 2a).

Participants were instructed to walk barefoot at a self-selected speed along an 8 m straight walkway, avoiding turns, on a smooth and sufficiently wide surface to prevent any obstruction, as shown in Figure 2b. The walking task was repeated four times—two trials per rater—with a five-minute rest period between repetitions, all conducted within a single session. No specific randomization or counterbalancing of trial order was applied, as all participants completed the same standardized task. Conducting all measurements on the same day and including rest periods minimized potential variability due to fatigue or day-to-day fluctuations, thereby ensuring consistent assessment of test–retest reliability. The trials were recorded simultaneously by the optoelectronic system BTS SMART DX and the Baiobit system. Subsequently, kinematic data acquired from BTS optoelectronic system were first normalized to the neutral position of each subject, obtained during a static test, then reconstructed and manually processed to extract the following seven spatiotemporal gait parameters:

Cadence [steps/min]: Stepping rate (1);(1)$$C=60\times \frac{s}{t}$$Stride length [m]: Distance between two consecutive foot falls at the moments of initial contacts (2);
(2)$$SL(k)=x_{{HS}_{r}}(k+1)-x_{{HS}_{r}}(k)$$Step length [m]: Distance between the initial contact of one foot and the initial contact of the opposite foot (3);
(3)$$STPL(k)=x_{{HS}_{r}}(k)-x_{{HS}_{L}}(k)$$Velocity [m/s]: Walking speed (4);
(4)$$V(k)=\frac{SL(k)}{GCT(k)}$$where$$GCT(k)={HS}_{r}(k+1)-{HS}_{r}(k)$$Swing phase [%GCT]: Average percentage of a gait cycle that either foot is off the ground (5);
(5)$$SW\left(k\right)=\frac{{HS}_{r}\left(k+1\right)-{TO}_{r}(k)}{GCT(k)}\times 100$$Stance phase [%GCT]: Average percentage of a gait cycle that either foot is on the ground (6);
(6)$$ST\left(k\right)=\frac{{TO}_{r}(k)-{HS}_{r}(k)}{GCT(k)}\times 100$$Single support phase [%GCT]: Average percentage of a gait cycle spent with only one foot on the ground (7);

(7)$$SS\left(k\right)=\frac{{HS}_{r}\left(k\right)-{TO}_{L}(k)}{GCT(k)}\times 100$$where GCT is the gait cycle time; k is the number of gait cycles; ${TO}_{r}$ is the toe-off right; ${HS}_{r}$ is the heel-strike right; s are the steps; t is the time of walk; ${HS}_{L}$ is the heel-strike left; ${TO}_{L}$ is the toe-off left; $x_{{HS}_{r}}$ is the position of heel-strike right; $x_{{HS}_{L}}$ is the position of heel-strike left.

For each subject, parameters from the trials were averaged across both sides (right and left) to obtain mean values.

All spatiotemporal parameters for both the marker-based system (BTS SMART-DX 6000) and the IMU system (Baiobit) were automatically computed by the proprietary algorithms of each manufacturer’s software. No custom procedures or additional algorithmic steps were applied by the authors. The systems provide direct outputs of cadence, velocity, stride length, step length, stance, swing, and single support phases, which were used for analysis. Because the comparison focused exclusively on derived spatiotemporal parameters, which are computed as intra-signal temporal differences, hardware synchronization between systems was not required. Both systems were activated simultaneously by the assessor, and no post hoc temporal alignment was applied.

### 2.4. Statistical Analysis

To assess the inter- and intra-rater reliability and validity between the two measurement systems, the following statistical methods were carried out: ICC, Standard Error of Measurement (SEM), Minimum Detectable Change (MDC), Spearman correlation analysis, two-tailed paired tests, Passing–Bablok (PB) linear regression, and Bland–Altman (BA) analysis.

The ICC analysis assess consistency of repeated measurements by the same rater (intra-rater, one-way) and between raters (inter-rater, two-way), estimating variability due to subjects versus measurements. ICC values range from 0 to 1 and were interpreted as follows: <0.40 (poor), 0.40–0.75 (moderate), >0.75 (good), and >0.90 (excellent) [35]. In the present study, intra-rater reliability was assessed using a one-way random-effects model (ICC(1,1), single measures, absolute agreement), as each participant was measured independently by the same rater. Inter-rater reliability was assessed using a two-way random-effects model (ICC(2,1), single measures, absolute agreement), considering the raters as representative of a larger population. The ICC(1,1) was calculated as follows (8):(8)$$ICC(1,1)=\frac{{MS}_{B}-{MS}_{W}}{{MS}_{B}+(k_{1}-1){MS}_{W}}$$

The ICC(2,1) was calculated as follows (8):(9)$$ICC(2,1)=\frac{{MS}_{B}-{MS}_{E}}{{MS}_{B}+\left(k_{2}-1\right){MS}_{E}+k_{2}\frac{{MS}_{R}-{MS}_{E}}{n}}$$
where ${MS}_{B}$ is the mean square between subjects; ${MS}_{W}$ is the mean square within subjects; $k_{1}$ is the number of measurements per subject; ${MS}_{E}$ is the mean square error; ${MS}_{R}$ is the mean square between raters; n is the number of subjects; and $k_{2}$ is the number of raters.

Spearman correlation analysis was carried out to assess the relationship between two measurement systems. Previously, a normality test based on Shapiro–Wilk test was carried out to choose the correct parametric (Pearson correlation) or not-parametric (Spearman correlation) analysis. Each correlation coefficient ranges from −1 to 1 and was interpreted as follows: 0.00–0.20 (weak), 0.30–0.50 (moderate), 0.60–0.70 (strong), >0.80 (very strong) [36]. In the present study, all parameters violated the assumption of normality; therefore, Spearman correlation was used for all analyses. The Spearman correlation coefficient (r) was calculated as follows (10):(10)$$r=\frac{cov(R_{X},R_{Y})}{\sigma_{R_{X}}\sigma_{R_{Y}}}$$
where $R_{X}$ is the rank of measurement X; $R_{Y}$ is the rank of measurement Y; $\sigma_{R_{X}}$ is the standard deviation of the rank $R_{X}$; $\sigma_{R_{Y}}$ is the standard deviation of the rank $R_{Y}$.

PB linear regression analysis was used to evaluate agreement between the two systems by identifying the presence of systematic error (either constant or proportional). A constant systematic error occurs when the 95% confidence interval (CI) for the intercept of the regression line (q) did not include 0 value. Similarly, a proportional systematic error occurs when the 95% CI for the slope of the regression line (m) did not include one value [37]. The slope (m) of the regression line was calculated as follows (11):(11)$$m=median(m_{ij})$$
where$$m_{ij}=\frac{y_{j}-y_{i}}{x_{j}-x_{i}}\;\mathrm{for}\;x_{j}\neq x_{i}$$

The 95% CI of the slope of the regression line was calculated as follows (12):(12)$$m_{95\%CI}=[m_{L},m_{U}]$$
where$$L=\frac{N-C_{0.05}}{2}\;\;U=\frac{N+C_{0.05}}{2}$$$$C_{\alpha }=1.96\sqrt{\frac{n(n-1)(2n-5)}{18}}\;\;N=\frac{n(n-1)}{2}$$

The intercept (q) of the regression line was calculated as follows (13):(13)$$q=median(y_{i}-mx_{i})$$

The 95% CI of the intercept of the regression line was calculated as follows (14):(14)$$q_{95\%CI}=[q_{L},q_{U}]$$
where$$q_{L}=median\left(y_{i}-m_{L}x_{i}\right)\;\;q_{U}=median(y_{i}-m_{U}x_{i})$$
where N is the number of valid slope estimates; $x_{i}$ is the i-th measure of the measurement X; $y_{i}$ is the i-th measure of the measurement Y; and $C_{\alpha }$ is the rank indices for the 95% CI of m.

The BA analysis was carried out to assess the agreement between the two systems and to evaluate discrepancies in reliability, either between raters or between sessions. The mean of the difference vector (bias), the 95% CI of the bias, and the limits of agreement (LOA) were calculated. The limits of agreement were calculated as follows (15):(15)$$LOA=bias\pm 1.96\times {SD}_{bias}$$

The 95% CI of bias were calculated as follows (16):(16)95%CIbias=bias±1.96×SDbiasn
where ${SD}_{bias}$ is a standard deviation of the bias; n is the number of measurements.

The distribution of the data points in the BA plot helps to identify systematic errors (both constant and proportional). A random distribution around the 0 line implies agreement between the two methods, while the presence of trends is indicative of the presence of errors (i.e., systematic proportional errors if the trend is linear) [38].

The SEM was calculated to quantify the variability in repeated measurements of each spatiotemporal parameter, being the representation of the magnitude of the variation between repeated measurements taken by the same individual [39]. It was computed using the following formula (17):(17)$$SEM=\sqrt{RMS}\times \sqrt{1-ICC}$$
where RMS is the Root mean square.

The MDC was also computed for each spatiotemporal parameter, representing the minimum magnitude of change that can be considered clinically relevant [40]. It was calculated as follows (18):(18)$$MCD=SEM\times 1.65\times \sqrt{2}$$

Finally, for two-tailed paired tests, a Shapiro–Wilk normality test was performed to evaluate the normality of each spatiotemporal parameter in order to choose the correct parametric (*t*-test) or non-parametric (Wilcoxon test) test.

In all the performed analyses, the uncertainty level was set at α = 0.05. All statistical analyses were performed using the RStudio software (version 2024.12.1+563, RStudio Team, Boston, MA USA) and MATLAB software (version 2025a, Mathworks Inc., Natick, MA, USA).

## 3. Results

### 3.1. Intra-Rater Reliability

Table 2 shows the intra-rater reliability analysis of spatiotemporal gait parameters measured using the Baiobit and BTS systems. The reliability analysis was conducted separately for each system by the same rater (Rater A for Baiobit and Rater B for BTS SMART-DX 6000) across two repeated measurement sessions. For each spatiotemporal parameter, Table 2 reports the mean and SD of the first and second measurements, the ICC, the SEM, and the MDC at 90% confidence. In addition, Table 2 presents the results of the BA analysis, including the bias, the 95% CI of the bias, and the LOA. Complementary reliability results, including the 95% CI of ICC values and the standard deviation of the mean difference vector (bias) from the BA analysis, are provided in Appendix A.

For cadence, Baiobit showed good reliability, with an ICC of 0.80 (Table 2). The SEM and MDC values were low (0.58 and 1.35, respectively), confirming measurement stability. The BA analysis showed a bias of −3.72, with its 95% CI (−6.56 to −0.88) that excludes 0, underlining the presence of a slight constant systematic error. Nevertheless, given the moderate bias magnitude, overall intra-rater reliability for cadence was adequate.

Regarding velocity, Baiobit demonstrated excellent reliability (ICC = 0.90, Table 2). The SEM and MDC values were very low (0.02 and 0.05, respectively), indicating precision in repeated measures. The BA analysis showed a negligible bias (0.05) with its 95% CI (−0.04 to 0.03) that includes 0, confirming the absence of systematic error. Overall, intra-rater reliability for velocity was excellent.

For step length and stride length, moderate reliability was found (ICC = 0.68 and 0.69, respectively; Table 2). The SEM and MDC values were very low (0.03 and 0.06, respectively), reflecting acceptable precision. The BA analysis indicated small biases (step length: 0.02; stride length: 0.04), with their 95% CIs (step length: −0.01 to 0.06; stride length: −0.02 to 0.11) including 0, highlighting the absence of constant systematic error. Taken together, intra-rater reliability for spatial parameters was adequate.

Concerning stance and swing phases, both parameters showed moderate ICC values (0.63, Table 2). The SEM and MDC values were larger than for global parameters (stance phase: 0.49 and 1.13, respectively; swing phase: 1.28 and 2.99, respectively), reflecting lower measurement precision. The BA analysis showed negligible biases (stance phase: −0.06; swing phase: 0.07), with their 95% CIs (stance phase: −0.96 to 0.84; swing phase: −0.83 to 0.97) including 0, underlining the absence of constant systematic error. However, the wider LOAs indicated greater variability. Overall, intra-rater reliability for stance and swing was doubtful.

For single support, reliability was lower (ICC = 0.53, Table 2), with higher SEM and MDC values. The BA analysis showed a bias of 0.29 with its 95% CI (−0.69 to 1.28) including 0, indicating the absence of constant systematic error. Nevertheless, the relatively wide LOAs indicated poor precision. Thus, intra-rater reliability for single support was considered doubtful.

Table 3 summarizes the results of the intra-rater reliability between the BAIOBIT and BTS SMART-DX Systems for each spatiotemporal parameter.

### 3.2. Inter-Rater Reliability

Table 4 shows results of the inter-rater reliability analysis of spatiotemporal gait parameters measured using the Baiobit and BTS systems. The analysis was conducted separately by two raters (Rater A and Rater B) using each system under similar conditions. For each spatiotemporal parameter, Table 3 reports the mean and SD of the measurements collected by each rater, the ICC, the SEM, and the MDC at 90% confidence. Furthermore, Table 3 reports the results of the BA analysis, including the bias, the 95% CI of the bias, and the LOA. Complementary reliability results, including the 95% CI of ICC values and the standard deviation of the mean difference vector (bias) from the BA analysis, are provided in Appendix A.

For cadence, reliability was excellent (ICC = 0.88, Table 4). The SEM and MDC values were low (0.38 and 0.88, respectively), indicating consistent measurement between raters. The BA analysis showed a small negative bias (−1.37) with its 95% CI (−3.61 to 0.87) including 0, confirming the absence of constant systematic error. Thus, cadence showed excellent inter-rater reliability.

Regarding velocity, reliability was excellent (ICC = 0.87, Table 4). The SEM and MDC values were low (0.04 and 0.10, respectively). The BA analysis showed a negligible bias (−0.02) with its 95% CI (−0.05 to 0.02) including 0, indicating the absence of systematic errors. Inter-rater reliability for velocity was excellent.

For step length and stride length, both parameters showed good reliability (ICC = 0.82 and 0.83, respectively; Table 4). The SEM and MDC values were low (step length: 0.04 and 0.09, respectively; stride length: 0.03 and 0.07, respectively). The BA analysis showed near-zero biases (step length: 0.00; stride length: 0.01), with their 95% CIs (step length: −0.02 to 0.03; stride length: −0.04 to 0.05) including 0, indicating the absence of systematic errors. Therefore, inter-rater reliability for spatial parameters was good.

Concerning stance and swing phases, ICCs were lower (0.66, Table 4), indicating moderate inter-rater reliability. The SEM and MDC values were higher compared to global gait parameters (stance phase: 0.46 and 1.08, respectively; swing phase: 0.47 and 1.09, respectively). The BA analysis showed small biases (stance phase: −0.42; swing phase: 0.41), with their 95% CIs (stance phase: −1.07 to 0.23; swing phase: −0.24 to 1.07) including 0, indicating the absence of systematic errors. Thus, reliability for stance and swing phases was from doubtful to good.

For the single support phase, reliability was moderate (ICC = 0.71, Table 4), with slightly elevated SEM and MDC values (0.36 and 0.83, respectively). The BA analysis revealed a small bias (0.12), with its 95% CI (−0.50 to 0.73) including 0, confirming the absence of constant systematic error, though with relatively broad LOAs. These results indicate from doubtful to good inter-rater reliability for the single support phase.

Table 5 summarizes the results of the inter-rater reliability between the BAIOBIT and BTS SMART-DX Systems for each spatiotemporal parameter.

### 3.3. Concurrent Validity

The results of the validity analysis are presented in Table 4 and Table 6. Table 6 shows the results of the two-tailed paired tests for each spatiotemporal parameter computed using the two systems, reported as mean ± SD, along with the results of the Spearman correlation analysis. Table 7 reports the PB linear regression analysis results for each spatiotemporal parameter, including m, q, and their respective 95% CI. Additionally, Table 7 reports the results of the BA analysis, including the bias, the 95% CI of the bias, and the LOA.

Figure 3, Figure 4, Figure 5, Figure 6, Figure 7, Figure 8 and Figure 9 show the PB and BA plots for all estimated spatiotemporal parameters.

Concerning cadence, the paired test did not show a statistically significant difference between the two systems (Table 6, *p*-value = 0.44), indicating overall agreement. This result was confirmed by the Spearman correlation (r = 0.88) and the BA analysis, where the 95% CI of the bias (−1.69 to 3.92) included the 0 value, excluding the presence of a constant systematic error (Table 7, Figure 3a). Furthermore, the random distribution of the points around the zero line suggested the absence of a proportional systematic error (Figure 3a). The PB analysis confirmed these findings, since the slope m was 1.07 and its 95% CI (0.78 to 1.44) included the reference value of 1, and the 95% CI of the intercept q (−50.50 to 28.52) included 0. Overall, for cadence, a good agreement between Baiobit system and the BTS system was found.

Regarding velocity, a statistically significant difference was found by the paired test, with Baiobit underestimating this parameter compared to the BTS system (Table 6, *p*-value < 0.001). The BA analysis confirmed the presence of a constant although minimal systematic error, as the bias was –0.06 and its 95% CI (−0.09 to −0.02) did not include 0 value (Table 7, Figure 4a). This result was also supported by the Spearman correlation showing a weak correlation between two systems (r = 0.23). The PB analysis indicated no proportional systematic error, as the slope m was 1.15 and its 95% CI (0.95–1.40) included 1. However, a near-constant systematic error was found, since the intercept q was −0.26 and its 95% CI (−0.60 to 0.01) was borderline. These findings suggest that, although velocity is slightly underestimated, the agreement between systems is acceptable.

For step length and stride length, the paired tests revealed significant differences, with Baiobit overestimating both parameters compared to BTS (Table 6, *p*-value < 0.001), although Spearman correlation showed good correlation (step length: r = 0.73, stride length: r = 0.70). BA analysis confirmed the presence of a constant systematic error, as the 95% CI of the biases did not include 0 (Table 7, Figure 5a and Figure 6a). Moreover, the PB regression confirmed the presence of proportional systematic errors, since the 95% CI of slopes m (step length: m = 1.84, 95% CI = 1.40–2.62; stride length: m = 1.74, 95% CI = 1.33–2.45) did not include 1. Consequently, no agreement was found between the two systems for step length and stride length parameters.

Concerning stance and swing phases, both parameters showed significant differences in the paired tests, with Baiobit underestimating stance phase and overestimating swing phase (Table 6, *p*-value = 0.02). BA analysis confirmed the presence of constant systematic errors, since the 95% CI of the biases (stance phase: 95% CI = −2.95 to 0.05; swing phase: 95% CI = −0.07 to 2.95) excluded 0 (Table 7, Figure 7a and Figure 8a). In addition, PB regression highlighted proportional errors, as the 95% CI slopes m (stance phase: m = 0.64, 95% CI = 0.25–1.34; swing phase: m = 0.65, 95% CI = 0.25–1.35) did not include 1. Both constant and proportional systematic errors were supported by the Spearman correlation which showed a very weak correlation (stance phase: r = 0.11, swing phase: 0.08). Taken together, these findings indicate that no agreement was observed between the two systems for stance and swing phases, which is expected given their complementary relationship.

Similarly, for single support phase, a statistically significant difference was found, with Baiobit overestimating this parameter compared to BTS (Table 6, *p*-value = 0.03). The BA analysis indicated the presence of a constant systematic error, as the bias was 1.43 and its 95% CI (−0.10 to 2.97) excluded 0 value, while PB regression confirmed the presence of a proportional systematic error, as the slope m was 0.68 and its 95% CI (0.26–1.48) excluded 1 (Table 8, Figure 9a,b). Spearman correlation supported the results (r = 0.11). These findings indicate that no agreement between the two systems was observed for single support phase.

Table 8 summarizes the results of the agreement between the BAIOBIT and BTS SMART-DX Systems for each spatiotemporal parameter.

## 4. Discussion

The current study aimed to assess the intra- and inter-rater reliability and the concurrent validity between Baiobit inertial system and BTS SMART-DX 6000 system in estimating spatiotemporal parameters in healthy subjects.

The results of intra-rater reliability of Baiobit system ranged from moderate (ICC = 0.61) to excellent (ICC = 0.91). Additionally, both systems showed low SEM and MDC values, indicating good accuracy and ability to detect real changes in outcome measures. These results are consistent with Bailo et al. [11], who reported good to excellent test–retest reliability (ICC > 85) for nearly all spatiotemporal parameters using a single IMU placed on the trunk. Similarly, other test–retest reliability studies showed high reliability for variables such as velocity and cadence (ICC = 0.77–1.00) using the same sensor placement [41,42].

For the inter-rater reliability, Baiobit system showed moderate to good reliability (ICC = 0.66–0.88). Velocity and cadence were the most reliable parameters, showing good to excellent reliability (ICC = 0.77–0.88). These findings align with previous studies investigating the test–retest reliability of these variables with a single sensor placed on the trunk [11,41,42]. By contrast, accuracy was reduced for stance, swing, and single support phases estimation, likely because a single trunk-mounted sensor cannot capture localized foot-ground contact events with sufficient precision. Previous studies have similarly reported that gait phases estimation requires either multiple sensors or algorithms specifically designed to estimate heel strike and toe-off from trunk kinematics.

Compared with other systems, the BTS system demonstrated greater overall stability, with ICC > 0.71 for 88% of the gait parameters, confirming its status as the “gold standard”. Guo et al. [29] similarly reported good to excellent reliability of BTS, with ICC values generally higher than those observed in the current study (>0.75). Differences may reflect variations in assessment procedures.

The validity analysis highlighted a clear distinction between global gait parameters, spatial metrics, and temporal gait phases. Among the global gait parameters, cadence showed the strongest agreement between Baiobit and the BTS system, with no significant difference (*p*-value = 0.44), a very strong correlation (r = 0.88), and no statistically significant bias with its 95% CI (−1.69 to 3.92) including 0 value. Velocity also demonstrated good validity, though Baiobit slightly underestimated it (*p*-value < 0.001). The BA analysis revealed a small but statistically significant bias (−0.06; 95% CI: −0.09 to −0.02). In contrast, the Passing–Bablok regression indicated no significant proportional bias, with the 95% confidence interval for the slope including 1 (0.95 to 1.40), confirming suitability for capturing overall gait metrics. These results are consistent with prior studies validating trunk-mounted IMUs for cadence and velocity [9,26,41].

By contrast, spatial parameters such as step and stride length showed systematic overestimation. Baiobit reported higher values for both step and stride length (*p*-values < 0.001), with consistent biases (0.06 and 0.11). Regression confirmed proportional errors, indicating the overestimation was systematic rather than random. This limits interchangeability with gold-standard systems, reflecting prior findings that single-sensor setups often misestimate spatial parameters unless supplemented by multiple sensors or alternative placements [26,28].

Finally, gait phase parameters (stance, swing, and single support) consistently showed poor validity. Stance phase was underestimated (*p*-value = 0.02), while swing (*p*-value = 0.02) and single support phases (*p*-value = 0.03) were overestimated. Although biases were small (around ±1.4%), proportional errors were present in all phases. These findings reflect the difficulty of a lumbar-mounted IMU in detecting foot-ground contact events critical for phase segmentation. Previous research emphasizes that accurate temporal analysis requires either sensors on the lower limbs or refined event-detection algorithms [23,26,41]. Overall, this study demonstrated that the Baiobit system achieves moderate to excellent reliability and good validity for global gait parameters in healthy adults. Moreover, it showed poor performance in detecting gait phase metrics (stance, swing, single support) and overestimation of spatial parameters (step and stride length), reflecting limitations specific to this system.

Previous studies using a single accelerometer placed on the trunk reported ICCs similar to ours [29,43,44,45,46]. For instance, Lim et al. [45] and Zijlstra et al. [46] showed moderate to high Pearson coefficients (r = 0.777–0.995) for velocity, cadence, stride, and step length. De Ridder et al. [29] compared Baoibit system with the GAITRite system finding high validity for velocity, cadence, and stride length (ICC 0.88–0.99), but low validity for support and swing time (ICC = 0.12–0.40). Bugané et al. [44] also observed poor agreement for estimation of gait phase parameters and no clear correlation between sensors at the foot and at L5. In contrast, McCamley et al. [47] achieved high accuracy in detecting initial and final foot contact using a sensor placed on the trunk, with mean errors of 0.02 ± 0.02 s and 0.03 ± 0.03 s, respectively. This contrasts with our study, where parameters dependent on initial and final foot contact showed high variability, suggesting lower accuracy of the system used in the present study. Mobbs et al. [48] reported that inertial sensors are useful for detecting global spatiotemporal parameters but limited in identifying subtle gait patterns differences. Similarly, Petraglia et al. [49] found moderate to low agreement in measurements of variability and asymmetry, while mean gait characteristics showed excellent agreement. Prisco et al. [21] further confirmed these findings, reporting good agreement for cadence and velocity but only moderate to poor agreement for gait cycle phase parameters. Together, these studies underscore the challenges of single lumbar-mounted IMUs in capturing temporal aspects of gait compared to gold-standard systems. A lumbar-mounted IMU system such as Baiobit also holds potential for clinical translation. Its portability, low cost, and minimal setup make it suitable for routine contexts where quick gait assessment is needed. In these scenarios, global gait parameters like velocity and cadence, which showed good validity in this study, may be sufficient to guide clinical decisions or identify patients who require more detailed evaluation. However, optical motion-capture systems remain preferable when high-precision spatiotemporal event detection or detailed segmental kinematics are required. Thus, single-sensor IMUs should be viewed as complementary tools that can expand access to gait assessment but do not replace laboratory-grade systems for complex analyses.

This study has several limitations. First, only two repeated measurements per subject were performed, which may have reduced statistical power and limited the robustness of reliability estimates. A slight discrepancy in intra-rater reliability between raters was also observed, potentially attributable to minor differences in calibration procedures or system handling. The relatively small sample size (*n* = 30), although generally adequate for paired statistical analyses and regression models, remains suboptimal for Bland–Altman analysis, for which larger cohorts (approximately 50 participants) are typically recommended to improve precision. Moreover, the study population consisted exclusively of healthy young adults, which inherently limits the generalizability of our findings to clinical populations. While the Baiobit system demonstrated moderate to excellent reliability and good validity for global gait parameters in this normative cohort, caution is warranted when extrapolating these results to older adults or individuals with gait impairments, such as patients with Parkinson’s disease or stroke. Indeed, pathological gait patterns often present greater variability, asymmetry, and altered foot–ground contact timing, which may affect both reliability and validity metrics. Therefore, future studies should extend validation to larger and more heterogeneous cohorts, including older adults and clinical populations with impaired gait, to determine whether the Baiobit system can reliably capture spatiotemporal parameters in pathological conditions. Additionally, the performance of lumbar-mounted IMUs could potentially be improved through multi-sensor configurations, refined event-detection algorithms, or integration with advanced computational methods, such as machine learning, to enhance detection of foot–ground contact events and improve accuracy for temporal and spatial gait metrics. These approaches may increase both the validity and clinical applicability of the system, particularly in settings where 3D motion capture is not available.

## 5. Conclusions

This study presents an assessment of the reliability and validity of Baiobit inertial system, based on lower back-mounted IMU, compared with the gold standard 3D optoelectronic system BTS SMART-DX 6000 in estimating spatiotemporal parameters in healthy subjects. The findings indicate that the Baiobit system demonstrates moderate to excellent intra- and inter-rater reliability and good validity for global gait parameters such as cadence and velocity, which were consistently measured with high agreement. Spatial parameters (step and stride length) showed moderate reliability but systematic overestimation in validity analyses, while gait phase parameters (stance, swing, single support) exhibited only moderate reliability and clear proportional systematic errors, limiting their interchangeability with a gold-standard system. Overall, Baiobit is well suited for assessing global spatiotemporal features of gait in clinical and research contexts where 3D motion capture is not available, but caution is needed when interpreting cycle phase outcomes, for which multi-sensor configurations or advanced algorithms remain necessary.

## Figures and Tables

**Figure 1 sensors-25-07643-f001:**
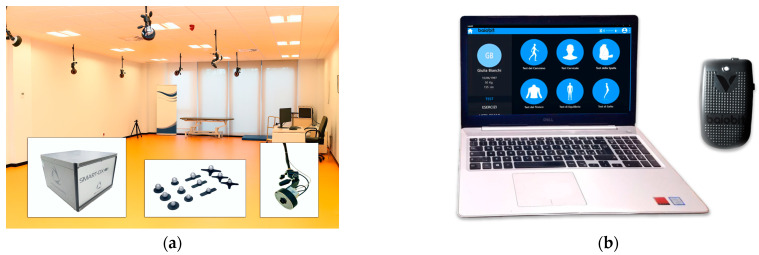
(**a**) 3D optoelectronic system BTS Gait Lab: movement analysis laboratory, retro-reflective passive markers and infrared-camera. (**b**) Baiobit system: inertial sensor and related software.

**Figure 2 sensors-25-07643-f002:**
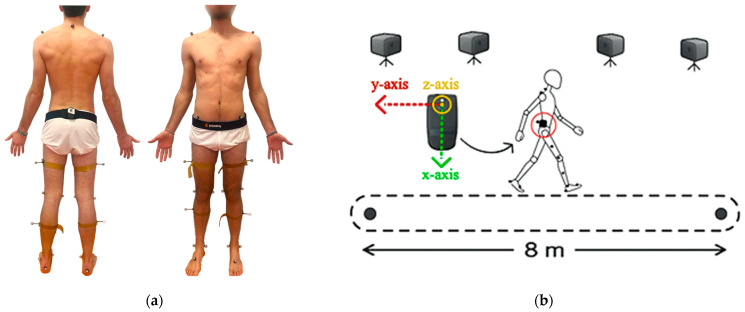
(**a**) Sensor and retro-reflective passive markers position. (**b**) 3D optical setup, IMU sensor coordinate orientation, walk protocol.

**Figure 3 sensors-25-07643-f003:**
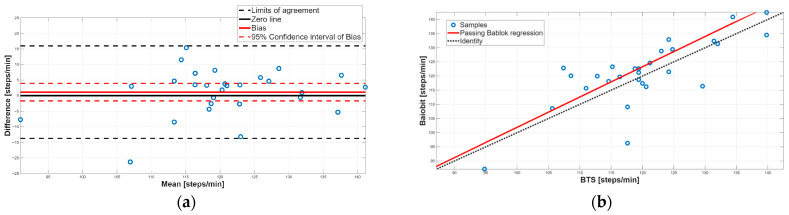
Cadence. (**a**) Bland–Altman plot with bias, zero-line, and limits of agreement. (**b**) Scatter plot with Passing–Bablok linear regression line and line of identity.

**Figure 4 sensors-25-07643-f004:**
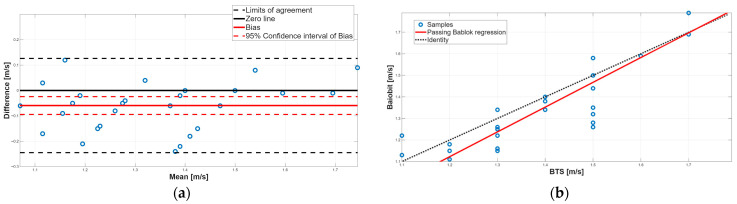
Velocity. (**a**) Bland–Altman plot with bias, zero-line, and limits of agreement. (**b**) Scatter plot with Passing–Bablok linear regression line and line of identity.

**Figure 5 sensors-25-07643-f005:**
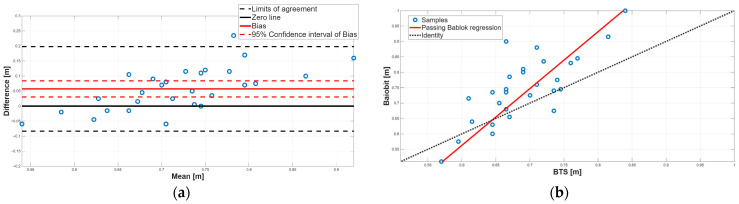
Step length. (**a**) Bland–Altman plot with bias, zero-line, and limits of agreement. (**b**) Scatter plot with Passing–Bablok linear regression line and line of identity.

**Figure 6 sensors-25-07643-f006:**
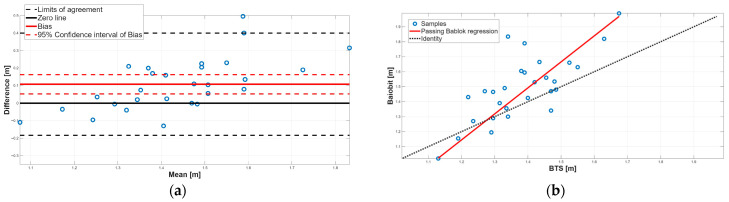
Stride length. (**a**) Bland–Altman plot with bias, zero-line, and limits of agreement. (**b**) Scatter plot with Passing–Bablok linear regression line and line of identity.

**Figure 7 sensors-25-07643-f007:**
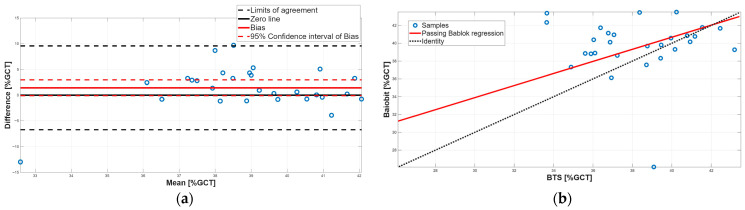
Single support phase. (**a**) Bland–Altman plot with bias, zero-line, and limits of agreement. (**b**) Scatter plot with Passing–Bablok linear regression line and line of identity.

**Figure 8 sensors-25-07643-f008:**
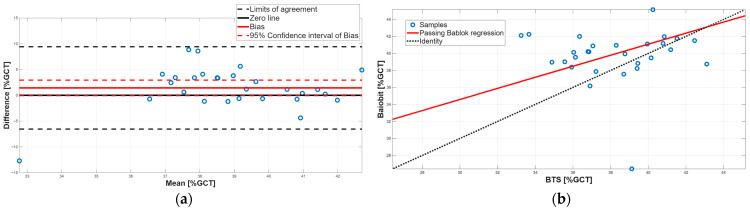
Swing phase. (**a**) Bland–Altman plot with bias, zero-line, and limits of agreement. (**b**) Scatter plot with Passing–Bablok linear regression line and line of identity.

**Figure 9 sensors-25-07643-f009:**
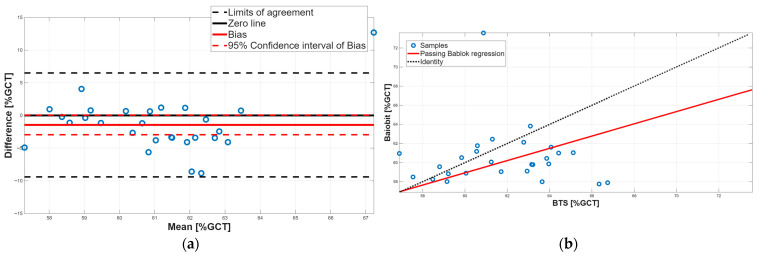
Stance phase. (**a**) Bland–Altman plot with bias, zero-line, and limits of agreement. (**b**) Scatter plot with Passing–Bablok linear regression line and line of identity.

**Table 1 sensors-25-07643-t001:** Demographic characteristics of study population.

Characteristics	Mean ± Standard Deviation
Age (years)	30.70 ± 14.69
Height (cm)	169.30 ± 7.74
Weight (kg)	68.83 ± 15.32

**Table 2 sensors-25-07643-t002:** Intra-rater reliability of spatiotemporal parameters measured by BAIOBIT and BTS SMART-DX Systems.

Intra-Rater Reliability—Baiobit System (Rater A)								
						Bland–Altman Analysis		
Spatiotemporal Parameters	1st Measure Mean ± SD	2nd Measure Mean ± SD	ICC	SEM	MDC	Bias	95% CI	LOA
Stance phase [%GCT]	60.31 ± 2.23	60.38 ± 3.13	0.63	0.49	1.13	−0.06	0.84 to −0.96	−4.70 to 4.58
Swing phase [%GCT]	37.31 ± 9.45	39.61 ± 3.13	0.63	1.28	2.99	0.07	0.97 to −0.83	−4.56 to 4.70
Single support phase [%GCT]	39.93 ± 2.00	39.64 ± 3.19	0.53	0.63	1.46	0.29	1.28 to −0.69	−4.79 to 5.38
Stride length [m]	1.53 ± 0.22	1.49 ± 0.21	0.69	0.03	0.06	0.04	0.11 to −0.02	−0.28 to 0.37
Step length [m]	0.77 ± 0.11	0.75 ± 0.11	0.68	0.03	0.06	0.02	0.06 to −0.01	−0.14 to 0.19
Velocity [m/s]	1.31 ± 0.20	1.32 ± 0.20	0.90	0.02	0.05	0.05	0.03 to −0.04	−0.18 to 0.17
Cadence [steps/min]	117.39 ± 13.91	121.10 ± 11.57	0.80	0.58	1.35	−3.72	−0.88 to −6.56	−18.36 to 10.92
**Intra-rater reliability—BTS system (Rater B)**								
						**Bland–Altman Analysis**		
**Spatiotemporal** **Parameters**	**1st Measure** **Mean ± SD**	**2nd Measure** **Mean ± SD**	**ICC**	**SEM**	**MDC**	**Bias**	**95% CI**	**LOA**
Stance phase [%GCT]	60.68 ± 1.73	60.68 ± 1.58	0.74	0.17	0.39	0.00	0.46 to −0.47	−2.39 to 2.39
Swing phase [%GCT]	43.49 ± 22.92	39.34 ± 1.59	0.74	1.98	4.62	−0.02	0.45 to −0.48	−2.40 to 2.37
Single support phase [%GCT]	39.86 ± 1.81	39.63 ± 1.83	0.74	0.07	0.17	0.22	0.74 to −0.29	−2.42 to 2.87
Stride length [m]	1.50 ± 0.18	1.53 ± 0.17	0.83	0.03	0.06	−0.03	0.01 to −0.07	−0.25 to 0.20
Step length [m]	0.75 ± 0.09	0.77 ± 0.12	0.85	0.03	0.07	−0.02	0.00 to −0.05	−0.14 to 0.09
Velocity [m/s]	1.33 ± 0.20	1.34 ± 0.20	0.88	0.03	0.07	−0.01	0.01 to −0.04	−0.18 to 0.15
Cadence [steps/min]	120.96 ± 9.23	120.24 ± 12.66	0.85	0.59	1.39	0.72	3.01 to −1.58	−11.12 to 12.54

Abbreviations: GCT = gait cycle time; SD = standard deviation; ICC = intraclass correlation coefficient; SEM = standard error of measurement; MDC = minimum detectable change; CI = confidence interval; LOA = limits of agreement.

**Table 3 sensors-25-07643-t003:** Results of the intra-rater reliability between the BAIOBIT and BTS SMART-DX Systems for each spatiotemporal parameter.

Spatiotemporal Parameters	Level of Intra-Rater Reliability
Stance phase [%GCT]	Doubtful
Swing phase [%GCT]	Doubtful
Single support phase [%GCT]	Doubtful
Stride length [m]	Adequate
Step length [m]	Adequate
Velocity [m/s]	Excellent
Cadence [steps/min]	Adequate

Abbreviations: GCT = gait cycle time.

**Table 4 sensors-25-07643-t004:** Inter-rater reliability of spatiotemporal parameters measured by BAIOBIT and BTS Systems.

Inter-Rater Reliability—Baiobit System								
						Bland–Altman Analysis		
Spatiotemporal Parameters	Rater A Mean ± SD	Rater B Mean ± SD	ICC	SEM	MDC	Bias	95% CI	LOA
Stance phase [%GCT]	60.28 ± 2.46	60.7 ± 1.57	0.66	0.46	1.08	−0.42	0.23 to −1.07	−3.73 to 2.89
Swing phase [%GCT]	39.72 ± 2.47	39.3 ± 1.57	0.66	0.47	1.09	0.41	1.07 to −0.24	−2.90 to 3.73
Single support phase [%GCT]	39.85 ± 2.35	39.73 ± 1.72	0.71	0.36	0.83	0.12	0.73 to −0.50	−2.98 to 3.22
Stride length [m]	1.51 ± 0.20	1.51 ± 0.19	0.83	0.04	0.09	0.01	0.05 to −0.04	−0.22 to 0.24
Step length [m]	0.76 ± 0.10	0.76 ± 0.09	0.82	0.03	0.07	0.00	0.03 to −0.02	−0.11 to 0.12
Velocity [m/s]	1.32 ± 0.20	1.33 ± 0.19	0.87	0.04	0.10	−0.02	0.02 to −0.05	−0.21 to 0.18
Cadence [steps/min]	119.33 ± 12.46	120.7 ± 10.84	0.88	0.38	0.88	−1.37	0.87 to −3.61	−12.71 to 9.97
**Inter-rater reliability—BTS system**								
						**Bland–Altman Analysis**		
**Spatiotemporal** **Parameters**	**Rater A** **Mean ± SD**	**Rater B** **Mean ± SD**	**ICC**	**SEM**	**MDC**	**Bias**	**95% CI**	**LOA**
Stance phase [%GCT]	61.90 ± 2.05	61.34 ± 2.23	0.71	0.19	0.45	0.56	1.17 to −0.05	−2.52 to 3.65
Swing phase [%GCT]	38.10 ± 2.05	38.64 ± 2.24	0.72	0.19	0.45	−0.55	0.06 to −1.15	−3.61 to 2.52
Single support phase [%GCT]	38.13 ± 2.05	38.80 ± 2.21	0.70	0.28	0.65	−0.68	−0.06 to −1.29	−3.77 to 2.42
Stride length [m]	1.38 ± 0.12	1.39 ± 0.13	0.83	0.03	0.06	−0.01	0.02 to −0.05	−0.19 to 0.14
Step length [m]	0.69 ± 0.06	0.69 ± 0.06	0.80	0.03	0.07	0.00	0.02 to −0.02	−0.08 to 0.07
Velocity [m/s]	1.37 ± 0.17	1.39 ± 0.18	0.83	0.02	0.09	−0.03	0.01 to −0.08	−0.21 to 0.15
Cadence [steps/min]	119.31 ± 9.39	120.81 ± 8.89	0.78	0.28	0.65	−1.50	0.83 to −3.83	−13.25 to 10.25

Abbreviations: GCT = gait cycle time; SD = standard deviation; ICC = intraclass correlation coefficient; SEM = standard error of measurement; MDC = minimum detectable change; CI = confidence interval; LOA = limits of agreement.

**Table 5 sensors-25-07643-t005:** Results of the intra-rater reliability between the BAIOBIT and BTS SMART-DX Systems for each spatiotemporal parameter.

Spatiotemporal Parameters	Level of Inter-Rater Reliability
Stance phase [%GCT]	Doubtful/Adequate
Swing phase [%GCT]	Doubtful/Adequate
Single support phase [%GCT]	Doubtful/Adequate
Stride length [m]	Adequate
Step length [m]	Adequate
Velocity [m/s]	Excellent
Cadence [steps/min]	Excellent

Abbreviations: GCT = gait cycle time.

**Table 6 sensors-25-07643-t006:** Two-tailed paired test and Spearman correlation analysis.

Spatiotemporal Parameters	Baiobit System Mean ± SD	BTS System Mean ± SD	Two-Tailed Paired Test (*p*-Value)	Spearman Coefficient (r)
Stance phase [%GCT]	60.31 ± 3.17	61.76 ± 2.58	0.02	0.11
Swing phase [%GCT]	39.68 ± 3.17	38.24 ± 2.58	0.02	0.08
Single support phase [%GCT]	39.70 ± 3.24	38.26 ± 2.56	0.03	0.11
Stride length [m]	1.49 ± 0.22	1.38 ± 0.13	0.00	0.70
Step length [m]	0.75 ± 0.11	0.69 ± 0.06	0.00	0.73
Velocity [m/s]	1.32 ± 0.20	1.38 ± 0.18	0.00	0.23
Cadence [steps/min]	121.24 ± 11.76	120.13 ± 10.13	0.44	0.88

Abbreviations: GCT = gait cycle time; SD = standard deviation; r = spearman coefficient.

**Table 7 sensors-25-07643-t007:** Passing–Bablok linear regression analysis and Bland–Altman analysis.

	Passing-Bablok Regression Analysis				Bland–Altman Analysis		
Spatiotemporal Parameters	m	95% CI m	q	95% CI q	Bias	95% CI Bias	LOA
Stance phase [%GCT]	0.64	0.25 to 1.34	20.48	−22.26 to 44.25	−1.45	−2.95 to 0.05	−9.40 to 6.50
Swing phase [%GCT]	0.65	0.25 to 1.35	15.14	−12.53 to 30.51	1.44	−0.07 to 2.95	−6.55 to 9.42
Single support phase [%GCT]	0.68	0.26 to 1.48	13.56	−16.99 to 29.72	1.43	−0.10 to 2.97	−6.71 to 9.58
Stride length [m]	1.74	1.33 to 2.45	−0.94	−1.90 to −0.37	0.11	0.05 to 0.16	−0.18 to 0.40
Step length [m]	1.84	1.40 to 2.62	−0.54	−1.06 to −0.23	0.06	0.03 to 0.08	−0.08 to 0.20
Velocity [m/s]	1.15	0.95 to 1.40	−0.26	−0.60 to 0.01	−0.06	−0.09 to −0.02	−0.24 to 0.13
Cadence [steps/min]	1.07	0.78 to 1.44	−5.01	−50.50 to 28.52	1.11	−1.69 to 3.92	−13.73 to 15.96

Abbreviations: GCT = gait cycle time; m = slope of the Passing–Bablok regression line; q = intercept of the Passing–Bablok regression line; CI = confidence interval of slope; LOA = limits of agreement.

**Table 8 sensors-25-07643-t008:** Results of the agreement between the BAIOBIT and BTS SMART-DX Systems for each spatiotemporal parameter.

Spatiotemporal Parameters	Level of Agreement	Type of Error
Stance phase [%GCT]	No Agreement	Constant and Proportional systematic errors
Swing phase [%GCT]	No Agreement	Constant and Proportional systematic errors
Single support phase [%GCT]	No Agreement	Constant and Proportional systematic errors
Stride length [m]	No Agreement	Constant and Proportional systematic errors
Step length [m]	No Agreement	Constant and Proportional systematic errors
Velocity [m/s]	Agreement	None
Cadence [steps/min]	Agreement	None

Abbreviations: GCT = gait cycle time.

## Data Availability

The datasets generated and analyzed during the current study are openly available in Zenodo at https://doi.org/10.5281/zenodo.17080275.

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
