# Peer review of "Assessing the Validity and Reliability of a Single Lumbar-Mounted IMU System for Gait Analysis"

_sensors, 2025, doi:10.3390/s25247643_

Round 1
Reviewer 1 Report
Comments and Suggestions for Authors
This is a well written paper assessing the reliability and validity of a commercial IMU device that extracts spatio-temporal parameters.
However, there are several aspects that need attention. It is unclear what the rol of the two raters (A and B) are; considering that a third rater (C) was responsible for data collection. Moreover, in order to assess the results, more information is needed on the method of extracting spatio-temporal parameters from the marker based system. No information is given on the methods for extracting spatio-temporal parameters from the IMU system. All this is necessary information to be able to replicate this study. and assess the systems used.
You only include healthy adults in this study, making the added clinical relevance of this study low. Based on these results clinicians will not be able to discern if this tool could be used for any population other than "healthy adult". So any claims that this tool could potentially be used in a clinical setting are unfounded. This also severely limites the added value of this work to the vast body of work in validation and reliability studies from lower trunk IMUs.
In the discussion you mention "a fundamental limitation of single lumbar-mounted IMUs" (line 446). As you have not shown any analysis method to extract the spatio-temporal parameters that claim is unfounded. You can only claim difficulties with this specific sensor and (I'm assuming) the proprietary software.
Reviewer 2 Report
Comments and Suggestions for Authors
Dear Authors,
Thank you for the opportunity to review your manuscript, “Assessing the Validity and Reliability of a Single Lumbar-Mounted IMU System for Gait Analysis.” This is a well-conceived and carefully executed study that contributes to the growing evidence supporting pragmatic, clinic-friendly IMU solutions for gait assessment. Your analytical approach (relative and absolute reliability, agreement analyses) is appropriate, and the manuscript is clearly written.
Below I provide minor, actionable suggestions to enhance transparency, reproducibility, and readability.
1) Title and Abstract
-
Title: Accurate and concise. No change required.
-
Abstract: Consider adding (i) the sample size, (ii) the key metrics compared (e.g., spatiotemporal variables, angular velocities if applicable), and (iii) one or two headline statistics (e.g., range of ICCs, mean bias with 95% LOA) to quantify your main findings. Avoid interpretive language in the abstract; keep it strictly descriptive.
2) Introduction
-
The background is thorough. To improve focus, you could condense early paragraphs and end with a clear, explicit hypothesis (e.g., “We hypothesized that the single lumbar IMU would demonstrate good-to-excellent agreement with the reference system for spatiotemporal gait variables and acceptable test–retest reliability.”).
-
Briefly state why a lumbar placement is clinically pragmatic (setup time, comfort) and which clinical scenarios might benefit most (primary care triage, rehab follow-up, community screening).
-
Important references about this topic has been omitted (doi: 10.3390/su12031222).
3) Methods (clarity and reproducibility)
Please add/clarify the following items:
Participants and Protocol
-
Specify whether all measurements were acquired in a single session or across multiple days (this affects interpretation of test–retest reliability).
-
Report randomization/counterbalancing of trial order (if applicable) and rest intervals to mitigate fatigue effects.
-
Detail footwear (shod vs. barefoot), walkway type/length, and turn handling (if trials involved out-and-back paths).
IMU Instrumentation
-
Provide sampling frequency, sensor range, filtering (e.g., filter type and cutoff), and axis orientation at the lumbar landmark (e.g., L5/S1). If any sensor calibration or hard/soft-iron compensation was performed, describe it briefly.
Reference System and Synchronization
-
Describe how the IMU and optical system data were time-aligned/synchronized (hardware trigger, software alignment by event detection, etc.). If no hard sync was used, clarify the post-hoc alignment approach.
Outcome Variables
-
List the primary and secondary gait variables compared (e.g., speed, cadence, step length, stride time, stance/swing time, variability metrics). Note any derived metrics and formulas.
Reliability and Agreement Analyses
-
Specify the ICC model and form (e.g., ICC(2,1) two-way random, absolute agreement; or ICC(3,k)), and the justification for the choice.
-
Provide the formulas or citations used for SEM and MDC (e.g., SEM = SD·√(1–ICC); MDC95 = 1.96·√2·SEM).
-
For Bland–Altman, indicate whether you tested proportional bias (regression of differences vs. means) and state the mean bias and 95% LOA explicitly in the text.
-
For Passing–Bablok, report the slope and intercept with 95% CIs and interpret whether constant and/or proportional bias is present.
-
If you assessed multiple variables, note whether you applied any multiplicity control or, at minimum, emphasize that interpretation focuses on pre-specified primary outcomes.
Data Handling
-
State how you handled missing/outlier trials (predefined rules), and confirm blinding procedures (e.g., raters processing IMU data blinded to optical results).
Ethics and Data Availability
-
Ensure the ethics approval identifier and consent procedures are explicit.
-
Consider a brief data/code availability note to bolster transparency.
4) Results (presentation and emphasis)
-
The results are well organized. To enhance readability:
-
Provide a brief “key findings” paragraph highlighting which variables showed good-to-excellent validity and reliability, and which were weaker.
-
In tables/figures, present ICC with 95% CI, SEM/MDC, and for agreement bias with 95% LOA. Where useful, add a compact clinical interpretation (e.g., whether MDC is smaller than typical clinical change).
-
Consider merging tables that report closely related outcomes to reduce redundancy.
-
If specificity allows, add complementary error metrics (e.g., MAE or RMSE) for intuitive interpretation.
-
5) Discussion (scope and clinical implications)
-
Your interpretation is balanced. Two refinements:
-
Clinical translation: Add 1–2 paragraphs on how a lumbar-mounted single IMU could replace or triage optical assessments in routine clinics (e.g., postoperative monitoring, fall-risk screening, telerehabilitation follow-ups), and where optical systems remain preferable.
-
Limitations and future work: You already note healthy adults and single-sensor configuration. Also mention generalizability to pathological gait, and suggest future studies with patient populations, dual-task walking, different speeds, and multi-sensor setups if aiming to capture segmental kinematics.
-
6) Figures and Tables
-
Ensure figures include axis labels, units, and (for Bland–Altman) clear depiction of mean bias and LOA. If proportional bias is present, consider overlaying the regression line.
-
Maintain consistent decimal precision and define all abbreviations in captions (ICC, LOA, SEM, MDC).
7) Writing and Style
-
The manuscript is well written. A light language polish could further tighten long sentences in the Introduction/Results.
-
Check uniformity of units (e.g., m/s, steps/min) and abbreviations across text, tables, and figures.
-
Verify reference formatting per the journal style; remove any redundant citations.
Overall Appraisal
This is a methodologically sound, clearly reported study that provides useful evidence on the validity and reliability of a single lumbar IMU for gait analysis in healthy adults. The requested edits are minor and aimed at sharpening reproducibility and practical interpretation. Once addressed, I would support acceptance.
Sincerely :)
Reviewer 3 Report
Comments and Suggestions for Authors
The manuscript presents an interesting study focused on evaluating the validity of wearable inertial systems, which are increasingly adopted in clinical practice. The paper emphasises both the growing applicability of these devices and their potential unreliability in estimating certain spatio-temporal parameters. Moreover, it highlights the importance of sensor placement and how positioning at specific anatomical landmarks may affect measurement accuracy.
The literature review is generally comprehensive and well-structured, providing an adequate background on the topic. The paper is well written and structured.
However, some methodological aspects require clarification or further detail to improve the scientific rigour and reproducibility of the study:
1. Intraclass Correlation Coefficient (ICC):
It is not specified which ICC type was used (e.g., ICC(2,1), ICC(3,1), etc.). Please indicate the model and the justification for its selection, as different ICC forms can yield distinct interpretations of reliability.
2. Protocol Description:
More information should be provided regarding the Hayes protocol—specifically, the number and placement of the markers used. This would help readers better understand the reference system and data acquisition process.
3. Statistical Analyses:
The rationale for choosing the Spearman correlation coefficient instead of the more conventional Pearson correlation should be clarified. Was a preliminary test for normality of the features conducted to justify the use of a non-parametric approach?
4. Statistical Consistency and Reporting:
The manuscript would benefit from a more uniform description of the statistical methods. It is recommended to report all relevant formulas and parameters to ensure transparency and reproducibility.
